# Acceptability and appropriateness of a risk-tailored organised melanoma screening program: Qualitative interviews with key informants

**Kate L. A. Dunlop**[1,2]*, **Louise A. Keogh**[3], **Andrea L. Smith**[1], **Sanchia Aranda**[4], **Joanne Aitken**[5], **Caroline G. Watts**[1,6], **Amelia K. Smit**[1,2], **Monika Janda**[7], **Graham J. Mann**[2,8,9], **Anne E. Cust**[1,2‡], **Nicole M. Rankin**[10‡]

1 The Daffodil Centre, The University of Sydney, A Joint Venture with Cancer Council NSW, Sydney, NSW, Australia, 2 Melanoma Institute Australia, The University of Sydney, Sydney, NSW, Australia, 3 Centre for Health Equity, Melbourne School of Population and Global Health, University of Melbourne, Melbourne, Victoria, Australia, 4 School of Health Sciences, University of Melbourne, Melbourne, Australia, 5 Viertel Cancer Research Centre, Cancer Council Queensland, Brisbane, Queensland, Australia, 6 Surveillance, Evaluation & Research Program, Kirby Institute, UNSW Sydney, Sydney, New South Wales, Australia, 7 Centre for Health Services Research, The University of Queensland, St Lucia, Queensland, Australia, 8 John Curtin School of Medical Research, Australian National University, Acton, Australian Capital Territory, Australia, 9 Centre for Cancer Research, Westmead Institute for Medical Research, The University of Sydney, Westmead, Sydney, New South Wales, Australia, 10 Centre for Health Policy, Melbourne School of Population and Global Health, University of Melbourne, Melbourne, Victoria, Australia

‡ AEC and NMR are joint senior authors on this work.
* kate.dunlop@sydney.edu.au

**Data Availability Statement:** Data cannot be shared publicly due to privacy or ethical restrictions. Public availability may compromise participant confidentiality. Reasonable requests for

# Abstract

## Introduction

In Australia, opportunistic screening (occurring as skin checks) for the early detection of melanoma is common, and overdiagnosis is a recognised concern. Risk-tailored cancer screening is an approach to cancer control that aims to provide personalised screening tailored to individual risk. This study aimed to explore the views of key informants in Australia on the acceptability and appropriateness of risk-tailored organised screening for melanoma, and to identify barriers, facilitators and strategies to inform potential future implementation. Acceptability and appropriateness are crucial, as successful implementation will require a change of practice for clinicians and consumers.

## Methods

This was a qualitative study using semi-structured interviews. Key informants were purposively selected to ensure expertise in melanoma early detection and screening, prioritising senior or executive perspectives. Consumers were expert representatives. Data were analysed deductively using the Tailored Implementation for Chronic Diseases (TICD) checklist.

## Results

Thirty-six participants were interviewed (10 policy makers; 9 consumers; 10 health professionals; 7 researchers). Key informants perceived risk-tailored screening for melanoma to

data may be sent to nicole.rankin@unimelb.edu.au or anne.cust@sydney.edu.au or alternatively The University of Sydney Human Research Ethics Committee, Sydney (human.ethics@sydney.edu.au: Project 2021/253).

**Funding:** This study received project grant funding awarded to AEC from the National Health and Medical Research Council (NHMRC #1165936) (NHMRC #2009923). AEC receives a NHMRC Investigator Grant (#2008454). KLAD receives an NHMRC Postgraduate Research Scholarship, NHMRC Supplementary Scholarship 2022 The University of Sydney and The Erik Mather PhD Scholarship (Melanoma Institute Australia). The funders had no role in study design, data collection and analysis, decision to publish, or preparation of the manuscript.

**Competing interests:** The authors have declared that no competing interests exist.

be acceptable and appropriate in principle. Barriers to implementation included lack of trial data, reluctance for low-risk groups to not screen, variable skill level in general practice, differing views on who to conduct screening tests, confusing public health messaging, and competing health costs. Key facilitators included the perceived opportunity to improve health equity and the potential cost-effectiveness of a risk-tailored screening approach. A range of implementation strategies were identified including strengthening the evidence for cost-effectiveness, engaging stakeholders, developing pathways for people at low risk, evaluating different risk assessment criteria and screening delivery models and targeted public messaging.

## Conclusion

Key informants were supportive in principle of risk-tailored melanoma screening, highlighting important next steps. Considerations around risk assessment, policy and modelling the costs of current verses future approaches will help inform possible future implementation of risk-tailored population screening for melanoma.

## Introduction

Early detection of melanoma is associated with better survival [1]. However, international evidence-based guidelines report there is insufficient evidence to recommend for or against routine screening for melanoma in the asymptomatic population [2] due to insufficient evidence that screening using current approaches reduces mortality [3, 4], concern about the potential harms associated with the overdiagnosis of slowly-progressive lesions [4–6] and that costs may outweigh benefits. Australia has the highest incidence of melanoma worldwide [7], and higher costs of treating late-stage melanoma add to the rapidly rising healthcare costs of melanoma [8]. Currently in Australia, skin checks primarily occur opportunistically, and are often consumer initiated but may also be offered by the clinician [9]. Higher socio-economic status has been associated with consumers who access skin checks [10]. The majority of skin checks are conducted by General Practitioners (GPs) in generalist practices or dedicated primary care skin cancer clinics [9], and they are also conducted by specialists in dermatology settings [11]. Melanoma overdiagnosis [5] is evidenced by the rising rate of in situ melanoma over time alongside minimal changes to mortality rates, and requires population-level interventions to minimise this harm [12].

A risk-tailored approach to screening for melanoma offers a promising way forward [3, 13–15] and melanoma risk prediction tools validated for the Australian population are available [16–18]. New technologies such as surveillance photography, teledermatology and artificial intelligence, have the potential to support clinicians in screening [19]. Risk-tailored screening, also referred to as 'risk-stratified screening' or 'targeted screening', tailors screening (e.g., eligibility, frequency, intervals, type of test) to individual risk rather than the mostly one-size-fits-all approach of organised population screening programs [20, 21]. Risk assessment for melanoma is complex and may include risk factors such as age, gender, family history, sun protection behaviours, sun exposure, skin type, number of naevi (moles) and genetic factors, including polygenic scores that combine the effects of many common genomic variants into a single risk score [16, 22, 23].

Moving from opportunistic screening to an organised risk-tailored approach to melanoma screening will be challenging as it will require changes to practice and policy [21, 24].

Nevertheless, several countries are moving forward with implementing risk-stratified screening as part of their long-term cancer plans [25–27]. The Australian Population Based Screening Framework outlines the key issues to be considered by decision-makers when assessing potential screening programs, and include criteria adapted from the World Health Organisation principles of screening as well as key principles for the implementation and management of screening programs [28]. A risk-tailored melanoma screening program would additionally require individual melanoma risk assessment and tailoring different aspects of the screening program to different risk levels [29]. This would include redistributing health service resources towards higher-risk individuals to screen more frequently whereas those at lower risk may screen less frequently or perhaps not all. Integrating risk factors such as lifestyle behaviours, environmental exposures and personal genomic risk information in addition to traditional risk factors to determine screening eligibility will also bring challenges in communication [30, 31]. These potentially significant policy and practice changes would have implications for the community, health professionals, organisations and health systems. Acceptability and the appropriateness of targeted screening for these stakeholders will be important for successful implementation [32].

Acceptability of risk-tailored screening programs by key stakeholders has been explored primarily in breast cancer reporting a high level of acceptability [33, 34] but with some reluctance to stop organised screening for those at low risk [35, 36]. To our knowledge, no studies have explored stakeholders' views of the acceptability and appropriateness of risk-tailored melanoma screening [37]. Pre-implementation research enables a multilevel approach alongside emerging evidence, guiding the future translation of complex innovations into practice that require health system change [38]. This approach will be timely while evidence for this new paradigm in melanoma screening is further developed. Qualitative interviews are ideally suited to exploring individual perspectives [39] and implementation frameworks ensure that implementation research is underpinned by relevant theory.

This study aimed to explore the views of key informants in Australia on the acceptability and appropriateness of risk-tailored organised screening for melanoma. Specifically, we aimed to identify barriers, facilitators and strategies to inform potential future successful implementation.

## Methods

### Participants and recruitment

Semi-structured interviews were conducted with key informants from four expert groups: policy makers; consumers with lived experience of melanoma and/or recognised community advocates; health professionals; and researchers. Study investigators developed a list of key informants with expertise and demonstrated interest in early detection and screening of melanoma in Australia. Key informants were included if they were recognised as a leader in their field, clinical or policy expert or experienced representative of melanoma patient or community groups. Potential participants were purposively selected from key national and state cancer and melanoma organisations, melanoma consumer organisations, clinical specialties of general practice and dermatology, the Australian Government Department of Health and Medical Research organisations. Participants were excluded if they did not hold a current position associated with the key organisations. Almost all participants held senior or executive positions and consumers had expert training as representatives of patients and communities.

Participants were invited by an email that included an invitation letter, information sheet, and consent form. Participant's contact emails were publicly available. At the completion of each interview, participants were asked to nominate individuals they believed should be

invited to take part in the study. Recruitment continued until no new information was generated (i.e., reached data saturation). This research was approved by The University of Sydney Human Research Ethics Committee (Project 2021/253).

## Implementation frameworks

This study used two implementation frameworks. Interview questions were informed by Proctor and colleague's *Outcomes for Implementation Research* framework to focus questions on implementation outcomes, in particular the concepts of acceptability (i.e., satisfaction with various aspects of the innovation) and appropriateness (i.e., perceived fit, relevance, compatibility, practicability) [32]. The Tailored Implementation for Chronic Diseases Checklist (TICD) was used to guide analysis in identifying the factors that influence implementation outcomes [40]. TICD is a meta-theoretical determinants framework with 57 potential determinants of practice grouped in seven domains, enabling barriers and facilitators to be matched to identify determinants and subsequently implementation strategies.

## Data collection

The semi-structured interview guide (included in S1 File) was piloted with two key informants and some items were consequently refined. Interview questions explored beliefs and perceptions about melanoma early detection and screening in Australia, the concept of risk-tailored screening, evidence for implementation and the key determinants (barriers and facilitators). Interviews were conducted between August 2021 and June 2022, and all were conducted by Zoom except for one by telephone, by one researcher (KD). Interviews were audio recorded. Participants were reimbursed for their time with a $80 AUD gift voucher.

## Data analysis

Interviews were transcribed verbatim, and identifying details removed. Data were analysed using the TICD checklist. Three investigators (NR, AC, KD) initially read four transcripts (one from each expert group) to familiarise themselves with the data and to guide selection of the optimal framework for analysis. Three frameworks were considered: the TICD checklist, Consolidated Framework for Implementation Research (CFIR), and EPIS Implementation Framework (EPIS). The TICD checklist was considered the best fit for a new health intervention in this context and 20% of transcripts were coded into the TICD checklist independently by two investigators (KD & NR) using a deductive approach. Codes and definitions of TICD determinants were then discussed with the wider investigator team and the coding framework was finalised. Remaining coding was completed by one investigator (KD), but was reviewed, discussed and revised based on ongoing feedback with the study team.

Findings related to acceptability and appropriateness of risk-tailored melanoma screening were coded under the seven domains of the TICD checklist: Guideline factors; Individual health professional factors; Patient factors; Professional factors; Incentives and resources; Capacity for organisational change; and Social, political and legal factors. The findings are coded to the most relevant TICD potential determinants of practice within each of these seven domains; some of the determinants were not relevant or infrequently discussed and these were omitted from our reporting. TICD domains and determinants relevant to this study are included in Fig 1.

Barriers and facilitators were identified within the coding framework. All investigators contributed to the final analysis and the study team reviewed and adjudicated any discrepancies around coding and identification of barriers and facilitators. Files were managed in NVivo 12

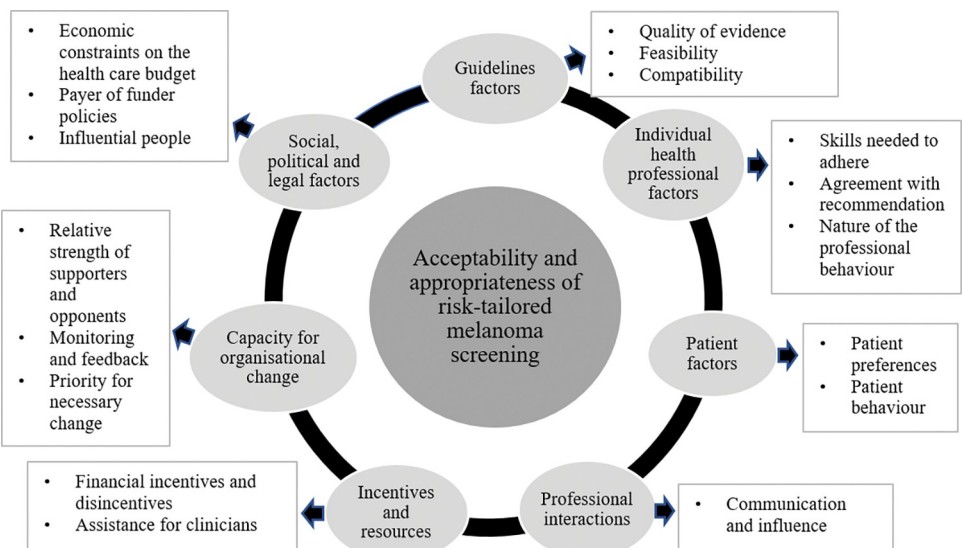

**Fig 1. TICD checklist [40] seven domains and seventeen determinants relevant to acceptability and appropriateness of risk-tailored melanoma screening.**

(QSR International, Australia) software. The conduct, design and reporting of this study follows the Consolidated Criteria for Reporting Qualitative Research (COREQ) [39].

## Results

We invited 47 key informants for an interview and 36 (83%) gave informed consent and took part. Twenty-eight participants had organisation-wide accountabilities (chief executive or director/head of an organisation or government department) or held similar roles previously, 12 were leading an academic or research program and 6 were founding directors or board director patient advocates. Interviews were 21–59 minutes (mean 30 minutes) in length. The demographic characteristics of the participants are summarised in Table 1.

The coding framework with TICD domains, relevant determinants, definitions and additional supporting quotes is included in S1 Table (Supporting information). Table 2 shows the barriers, facilitators and potential strategies for implementation identified within the coding framework.

Findings are presented below in respect to risk-tailored melanoma screening (targeted health intervention) as follows: Depiction of the current climate for early detection of melanoma; Considering risk-tailored melanoma screening; and Barriers and facilitators to implementation of risk-tailored melanoma screening.

### Depiction of the current climate for early detection of melanoma

**Early detection exists but is ad hoc and fragmented.** Participants across all groups described the current situation of early detection of melanoma in Australia as fragmented: *"disorganised is not correct, maybe unorganised is the best way to talk about it, in that it's ad hoc, it's dependent on the individual person and their GP or clinician." (P7, Policy Expert)*.

While one participant reported that *"we have a very good opportunistic screening system going on mostly in private practice by dermatologists and even more by so-called skin cancer doctors" (P31, Dermatologist)*, many participants, particularly GPs, expressed concern over the variable skill level related to skin cancer screening: *"I see very weak and highly variable quality*

Table 1. Demographic characteristics of participants.

| Characteristic | | Frequency (n) |
|---|---|---|
| **Expert Groups** | Policy makers and other interested experts | 10 |
| | Consumers and community advocates | 9 |
| | Health professionals<br>GPs (n = 5); dermatologists (n = 3); practice nurse (n = 1); other medical specialist (n = 1) | 10 |
| | Researchers with relevant expertise | 7 |
| **Years of experience[a]** | 1–9 years | 4 |
| | 10–19 years | 9 |
| | ≥20 years | 23 |
| **Gender** | Female | 25 |
| | Male | 11 |
| **Australian State/ Territory** | New South Wales | 14 |
| | Queensland | 9 |
| | Victoria | 10 |
| | Tasmania | 1 |
| | Western Australia | 1 |
| | Australian Capital Territory | 1 |

[a]Expertise related to early detection and screening of melanoma in Australia and/or to cancer control policy or research

*of care in general practice, ranging from the very best through to dreadful." (P34, GP)* This was attributed in part to the absence of comprehensive training for GPs in skin cancer and of an accreditation process related to skin cancer checks: *"any GP can call himself an expert and set up a clinic." (P22, GP)* (Domain 2: Skills needed to adhere)

**Current practice leads to inequities and overservicing.** Policy experts, dermatologists and researchers reported that the current system leads to inequity and unnecessary excisions: *"there is a lot of informal screening going on and the problem with informal screening is it tends to focus on the worried well and it can generate significant issues of inequity in a user-led screening process. We are all well aware of anecdotes of patients that are effectively over-screened and that leads to a lot of unnecessary interventions like biopsies." (P20, Policy Expert)*

Health professional and patient fear related to misdiagnosis were also seen to drive overservicing: *"we're going to have a big problem trying to stop the diagnosis of lesions that actually in hindsight didn't need to come off. . .no one wants to be the GP who has a patient die from melanoma when they saw that lesion six months ago." (P24, Researcher)* One dermatologist further explained *"she (consumer) got into a rant about the term unnecessary excisions because she was saying what doctors and public health people call unnecessary excisions, for the patients, it's not an unnecessary excision, it's basically peace of mind." (P31, Dermatologist)* (Domain 3: Patient preferences)

Furthermore, there was concern about the impact of overservicing on health costs and that the current practice rewards more frequent skin checks than is necessary: *"it seems like there are some quite strong commercial drivers for the way it's being delivered and perhaps that's helping drive the overdiagnosis of lesions that maybe wouldn't actually need to be detected." (P17, Researcher)* (Domain 6: Relative strength of supporters and opponents).

**Results in less-than-ideal health outcomes (melanoma).** Participants expressed concern that many people who may need screening were being missed: *"we have this sort of mixed*

**Table 2. Barriers, facilitators and potential strategies for implementation of risk-tailored melanoma screening linked to TICD domains and relevant potential determinants of practice.**

| TICD* Domains | TICD determinants | Barriers and facilitators | Potential implementation strategies |
|---|---|---|---|
| Guideline Factors | Quality of evidence supporting the recommendation | **Barrier:** Lack of feasibility of an RCT in context of current practice (ie cost and mortality benefit from screening is hard to demonstrate over a short time frame) **Facilitators:** Acceptability of modelling to demonstrate effectiveness, as has been the case for cervical cancer screening Cost-effectiveness is a government priority | Strengthen evidence base for cost-effectiveness |
| | Feasibility | **Facilitator:** Primary health care professionals particularly GPs are well placed (currently undertake opportunistic screening) | Develop risk-tailored screening pathway in conjunction with models of care Consult GPs and primary care in planning and development |
| | Compatibility | **Facilitator:** Screening as an opportunity for prevention | Link prevention strategies with screening program |
| Individual Health Professional Factors | Skills needed to adhere | **Barriers:** Variable skill level in identification of suspicious lesions in general practice Absence of accreditation for GPs working in skin cancer **Facilitator:** Education/training programs exist for upskilling GPs to conduct total body skin examinations | Upskilling of GPs and health professionals as a priority including risk assessment (application of risk algorithm) Establish standards |
| | Agreement with recommendation | **Barrier:** Some were reluctant to advise reduced or no screening for low risk **Facilitator:** Key informants agree with risk-tailored screening in principle Perceived opportunity to improve equity of outcomes | Develop an option or pathway for people at low risk Develop clear targeted public messaging and communication |
| | Nature of the (professional) behaviour | **Barrier:** Views differ within professionals and between groups on who is best placed to conduct screening **Facilitator:** Primary health care professionals are well placed | Evaluate different risk assessment criteria and models of screening delivery including referral pathways to dermatologists and surgeons |
| Patient factors | Patient preferences | **Barrier:** Consumers priorities may differ in harms/ benefits of screening to expert groups **Facilitator:** Consumers with lived experience are strong advocates | Develop clear targeted public messaging and communication Develop educational resources for advocacy organisations, patients and healthcare providers |
| | Patient behaviour | **Barriers:** Reluctance to reduce screening frequency or exclude those at low risk Confusing public messaging compared to existing screening program messaging Potential for a backlash (low-risk groups may feel they have missed out or opt out) | Develop an option or pathway for people at low risk Develop clear public messaging and communication Develop clear policy related to low risk |
| Professional Interactions | Communication and influence | **Barrier:** GPs may not adhere to screening protocols or criteria if not engaged (eg continuing high frequency of screening of those at low risk | Consult GPs and primary care in planning and development |
| Incentives and resources | Financial incentives and disincentives | **Barrier:** Absence of a Medicare item number **Facilitator:** Importance of a financial incentive for providers | Consult all stakeholders in program design Consider impact of a Medicare item number for screening |
| | Assistance for clinicians | **Facilitator:** Novel technologies such as Total Body Photography (TBP) may reduce workload Availability of risk assessment tools | Support research in novel technologies |
| Capacity for Organisational Change | Relative strength of supporters and opponents | **Barrier:** Potential for conflict of interest for GPs whose current practice rewards frequent skin checks **Facilitator:** Widespread tensions for change | Consult all stakeholders in program design |
| | Monitoring and feedback | **Barrier:** Lack of consensus around what is needed to evaluate program **Facilitator:** Existing screening programs have performance indicators that can inform evaluation measures Socio-demographic status of participants can be monitored to help determine equity | Establish consensus around what outcomes for program evaluation |
| | Priority for necessary change | **Barrier:** Primary prevention seen by a small number (two outliers) as a greater priority for government spending | Link prevention strategies with screening program Advocate with policy makers |

*(Continued)*

**Table 2.** (Continued)

| TICD* Domains | TICD determinants | Barriers and facilitators | Potential implementation strategies |
|---|---|---|---|
| Social, Political and Legal factors | Economic constraints on the health care budget | **Barrier:** Competing health care system costs to fund a national screening program<br>**Facilitator:** Potentially risk-tailored screening will save money and lives | Pitch cost savings and lives saved to Government |
| | Payer of funder policies | **Barrier:** Lack of agreement and coordination across government jurisdictions. | Consult all stakeholders in program design |
| | Influential people | **Barriers:** Lack of cohesion across health system and stakeholder views<br>Competing interests | Consult all stakeholders in program design |

*TICD: Tailored Implementation for Chronic Diseases [*Flottorp et al., implement Sci, 2013*] [27]

*pattern of opportunistic detection, quasi screening but very poorly targeted and a lot of people who are at high risk based on phenotypic factors who are not being identified. . .so it's a bit of [a] muddle."* (P30, *Researcher*).

**Considering risk-tailored melanoma screening.** There was widespread support from study participants in principle for tailored melanoma screening based on risk: *"If we are going to have any sort of screening, it does need to be risk-stratified. . .we don't want to be doing a big screening program on everybody, only to find more low-risk melanoma."* (P18, *Researcher*) It was considered *"the only way forward"* (P31, *Dermatologist*) in the current climate. (Domain 2: Agreement with recommendation).

Participants described how risk-tailored melanoma screening would be highly feasible in primary care and GPs in particular, were recognised as being well placed. One GP explained: *"the place to base risk assessment processes is in general practice because they're the people that deal with these things first and the risk assessment is not that difficult."* (P22, *GP*) The nature of melanoma also meant that risk-tailored screening was regarded as practical: *"[melanoma screening] doesn't have the barriers, like mechanical barriers of breast screen needed, that's infrastructure to do the screening. It doesn't have the yuck factor associated with bowel cancer, which also inhibits people to screen."* (P2, *Policy expert*) (Domain 1: Feasibility- the practicality of risk-tailored screening).

Risk-tailored screening programs were seen to fit with current practice and an opportunity to provide education around skin cancer prevention: *"one beauty of screening programs is that it's an absolute opportunity of linking to prevention, so one can do a lot of education and prevention as a part of screening."* (P24, *Researcher*) (Domain: Compatibility of risk-tailored screening with existing workflow).

However, the most commonly reported barrier across the interviews was the insufficient high-quality research evidence to support risk-tailored melanoma screening reducing mortality and the need to strengthen the evidence base prior to introducing a screening program. Many participants acknowledged the challenges of conducting a randomised controlled trial (RCT) to demonstrate a mortality reduction and a cost benefit *"the biggest challenge is that the highest level of evidence of screening programs comes from randomised controlled trials, so that is exceptionally difficult now to perform in the setting where there's already a lot of screening occurring outside of any formal programs."* (P20, *Policy expert*) Some participants further highlighted that *"convincing people that we need to potentially limit screening (in low-risk people) will require, I think, a strong evidence base."* (P24, *Researcher*) (Domain 1: Quality of evidence supporting the recommendation).

While consumer advocates saw great value in risk-tailored screening, their priority was for consumer guidance around screening for melanoma: *"I mean at least people know what they're*

*up against. At the moment many people don't know anything about getting skin checks. . . So, I think tailored screening is brilliant, I'm all for that." (P4, Consumer advocate)* (Domain 3: Patient preferences).

## Barriers and facilitators to implementation of risk-tailored melanoma screening

**Views on who is the most appropriate to conduct risk-tailored screening.** There were mixed views on who is best placed to conduct screening. Some suggested nurses might be appropriate: *"You can just even have the nurses doing the risk assessment–even patients themselves can probably apply the algorithm." (P16, GP)* Another GP expanded saying *"if you're doing that in an organised, systematic way on large numbers of people, that's not doctors' work. That's a clinical assistant. It doesn't need to be a nurse either, but it needs to be an appropriately trained person. . .I mean most people these days can buy an airline ticket online, so they can get their melanoma risk score." (P34, GP)* Dermatologists were clear that potential population-based screening was not their role: *"I work in private practice and am increasingly getting frustrated with referrals that are just for a skin check, whereas I don't think that's my role to provide and I try to push back a little bit on that." (P35, Dermatologist)* (Domain 2: Nature of the professional behaviour).

Additionally, the extent of opportunistic screening currently undertaken by GPs was recognised as an opportunity: *"What you don't want to do is overlap and duplicate stuff that's already happening (in primary care) both in terms of the checks that are being done on the patients, the skin check, but also in terms of the clinicians that are being used to do it." (P26, GP)* (Domain 2: Nature of the professional behaviour).

**Importance of getting GPs on board to optimise delivery of risk-based screening.** Participants noted the importance of consulting with primary care practitioners in planning to optimise the delivery of screening which may not have been fully recognised previously: *"it has to have the support of the professional community and we should learn the lesson from bowel cancer there. GPs did not support the national [bowel cancer] screening program and I think that slowed it down. So, you've got to have that–alongside clear messaging to the public." (P1, Policy expert)* (Domain 4: Communication influence).

One GP further explained why GPs and dermatologists need to be included in planning:*"There will still be people presenting with lesions ad hoc to their GP. . . or if a person's being screened for melanoma but other stuff is identified, there's got to be a mechanism to deal with those issues." (P26, GP)* (Domain 2; Nature of the professional behaviour).

**Importance of getting buy-in from all stakeholders.** Many participants highlighted the importance of taking into account all the stakeholder views to reduce unbalanced influence on outcomes e.g., *"I think if it was rolled out as part of a national initiative that had a steering committee which took into account the various stakeholders, patients, community representatives, government representatives, academic melanoma doctors, dermatologists, skin cancer doctors, GPs, so with government funding, I don't think there'd be too much of an issue regarding self-interest." (P23, Dermatologist)* One participant noted the importance of learning from existing programs: *"I think there is a lot that we need to learn from the science behind the national bowel screening program and then frankly, the capital, the politics behind it." (P8, Policy expert)* (Domain 7: Influential people; Domain 6: Relative strength of supporters and opponents).

**Reluctance to reduce screening frequency for those at low risk.** Reluctance to exclude those at low risk was related to the reality that even if deemed low risk, individuals still have a chance of developing melanoma. One consumer explained: *"I know I found it frustrating It's like, 'oh God, you're trying'–you're doing everything you can to look after your health, but you're*

*coming up against roadblocks.. just because you're low risk doesn't mean you're not going to get melanoma." (P13, Consumer)* There was also reluctance to reduce screening frequency for those at low risk who currently have regular skin checks: *"That's going to be the challenge too, that people like me are emotionally attached to it [screening]. . .it takes time for even low-risk candidates who have been exposed to accept their low risk and to get on with life." (P13, Consumer)* (Domain 3: Patient Behaviour).

**Getting the consumer message right.** Participants, especially policy experts and health professionals, expressed concern about the possible repercussions on skin cancer prevention behaviour of a message that advises some groups they no longer need to screen. One policy expert remarked on the importance of social marketing and getting the message right: *"So this is going to have to be absolutely laser focused to get the message across to the community or we will face (a) a backlash and (b) we will actually face worse outcomes because of the groups that are screened, that actually don't respond to cues to which they would respond today." (P8, Policy maker)* (Domain 3: Patient behaviour).

**Establishing appropriate funding incentives.** Financial incentives for providers were viewed as an important factor influencing acceptability when participants described funding for risk-tailored screening. As one participant said: *"it would need to be funded appropriately through Medicare [Australia's universal health insurance scheme] or some alternate means to recognise the work and the time involved." (P26 GP)* Funding for services subsidised by the Australian Government was considered unavoidable by most: *"GPs will need an item number [Medicare funded] if you want them to do it." (P10, Policy maker)* However, a small number of participants challenged this approach to incentives: *"We used to have many more item numbers for nurses [in primary care] and that actually wasn't a good thing because it meant they did the things that the item numbers generated rather than the things that were actually good care." (P37, Practice Nurse)* (Domain 4: Financial incentives and disincentives).

**Concerns about potential loss of income for general practitioners.** Some participants spoke about the potential for a conflict of interest related to a change in practice. The potential for loss of income for some GP practices primarily established to conduct skin checks was recognised as a challenge and may lead to some opposition in implementation: eg., *"I think change in the system will be difficult with some stakeholders, particularly the medical practitioners who derive a lot of benefit from seeing a lot of patients, because they'd perceive a loss of income and I think that's going to be difficult to navigate." (P35, Dermatologist)* However, one policy expert expressed less concern as private breast screening clinics currently work alongside the National Breast Screening program: *"But I know generally speaking the breast clinics can be brought into the tent, but there's always a little bit of squabbling along the edges." (P10, Policy expert)* (Domain 6: Relative strength of supporters and opponents).

**Demonstrating cost-effectiveness to the government.** Overwhelmingly, funding for a national screening program was seen as the responsibility of the Commonwealth government, however cost was considered a barrier for implementation: *"Biggest barrier is the health dollar." (P16, GP)* (Domain 7: Economic constraints on the health care budget).

Evidence about cost-effectiveness was seen as critical for implementation: *"I think things like cost-effectiveness and health resources utilisation benefits are also important to show and that would almost be the first thing to cross off." (P35, Dermatologist)* Most participants reported modelling studies as a good option for strengthening evidence: *"modelling studies with some very good assumptions and some good data that will be developing over the next few years–and will produce some really excellent data that with modelling will be able to give us a pretty good picture of what the benefit of screening would be." (P32, Researcher)* However, some were more cautious *"If we go down that path [modelling] as the proof of concept, we need to ensure that our data actually reflects that is what is delivered. Because models are models." (P8,*

*Policy expert)* One researcher noted modelling has previously guided cancer screening programs: *"the previous government [screening] committee seemed to be supportive of potentially being guided by modelling and I think the change in the cervical cancer screening happened based on modelling to a large extent." (P28, Researcher)* (Domain 1: Quality of evidence supporting the recommendation).

Demonstrating the cost-effectiveness of the program was considered a facilitator: *"there is already significant cost burdens to the health system and the way I think to pitch this to government is that not only is this going to save lives, but it's going to be cost saving." (P2, Policy expert)*.

**Structural issues for funding healthcare.** Participants acknowledged the challenges associated with the Australian healthcare funding system, jointly run by all jurisdictions–federal, state and territory, and the impact: *"It's always that challenge, the state versus federal and where you've got hospital services that are funded by state funding and you've got obviously all primary care services are funded federally through Medicare and how that overlap often occurs with a screening program is sometimes interesting." (P26, GP)* One participant explained the consequences of an absence of strategy in place in a recent change in a national screening program: *"With the renewal of cervical screening; there was no social marketing at a national level of any substance and the states and territories were left to pick that up. So if we're going to do this, we actually have to do it properly." (P8, Policy expert)* (Domain 7: Payer of funder policies).

**Availability of risk-based assessment tools and novel technologies.** Several researchers and GPs pointed out that risk assessment tools for melanoma specific to the Australian population were currently available and easy to use: *"I mean we have two well-validated, Australian risk prediction models for melanoma that are based on phenotypic factors that could be easily collected. And looking to the future, they could potentially be supplemented by polygenic risk scores as well." (P30, Researcher)* Some participants referred to the rapid development of novel technologies such as Total Body Photography (TBP) as having great potential to assist clinicians rather than replace them in practice in the near future: *"I do think eventually we will have sufficient evidence of some of these automated analyses of dermoscopy that will add something above and beyond GPs using their naked eye assessment that could contribute to improving the sort of diagnostic efficiency of a screening program." (P30, Researcher)* (Domain 5: Assistance for clinicians).

**Screening vs prevention as a priority.** Two researchers challenged whether screening was a priority over prevention for melanoma in our current health system with competing health costs: *"But for people at really very, very high risk, I'm not against having a screening program, I just think that in terms of the priorities, prevention, primary prevention, is really where we need the funding." (P18, Researcher)* (Domain 6: Priority for necessary change).

**The importance of monitoring outcomes.** There was clear recognition of the importance of monitoring clinical outcomes as well as the performance (e.g., uptake or quality of the program) of an organised screening program for melanoma. One researcher reported *"Well the clinical quality is the top thing, so are melanomas being picked up early and not being missed and are things that are not melanomas [being included as well]?" (P32, Researcher)* Similarly, another researcher noted melanoma thickness as a measure, and equity as an issue: *"I think across the population we would want to see the medium thickness of melanomas falling and being maintained at an acceptable level. . .then one could look at jurisdictions and different sociodemographic groups to determine that that's being achieved in an equitable way" (P24, Researcher)*. The patient experience was mentioned as a program measure: *"But I think around the patient perspective of the journey too, that is it relatively seamless from the perspective of the patient . . . if they get a positive diagnosis, do they then get linked into services?" (P10, Policy Expert)* (Domain 6: Monitoring and feedback).

## Discussion

There is increasing interest in risk-tailored screening with the aim of maximising benefits and reducing harms associated with population screening, and this is the first study to explore the views of key informants on this approach for melanoma. We identified barriers and facilitators at many levels related to changing screening practice from opportunistic to an organised, risk-tailored approach. Based on our analysis, we have identified potential strategies that may inform future steps towards potential implementation of risk-tailored screening for melanoma.

Key informants perceived risk-tailored screening to be highly acceptable, although some participants from each of the consumer, policy and clinical roles were reluctant to exclude individuals deemed to be at low risk from participation in a screening program. Our findings showed that key informants see risk-tailored melanoma screening as a logical way to improve health outcomes for the population compared to the current system and reduce health costs related to melanoma. Risk-tailored screening was perceived as appropriate i.e., feasible and relevant to the Australian setting, where melanoma incidence is high, and GPs already provide risk assessment and skin checks as part of routine care. Melanoma may be well-suited to risk-tailored screening that requires simple technology such as dermoscopy and a less complex infrastructure for establishment.

Studies in breast cancer [33, 35, 36] also found that risk-tailored screening was considered highly acceptable in principle, but that some people at low risk would be reluctant to reduce or stop screening. The net benefits and cost-effectiveness of risk-tailoring screening may not be realised if people at low risk continue to screen frequently, but this may not be surprising considering the high level of melanoma awareness in Australia and the positive messages associated with cancer screening and early detection [41]. In a previous study [42], we reported that people who were not getting regular skin checks would find reduced screening frequency acceptable because melanoma was visible on their skin and thus, they felt some personal control over early detection of melanoma. Developing an option or pathway for people at low-risk, particularly current screeners, is a potential strategy that may help to address concerns for those at low-risk.

The need to demonstrate evidence of mortality benefit before implementation of a population screening program was singled out for attention. A small number of trials have reported challenges with a population approach to melanoma screening. Germany's nationwide skin screening program, adopted following a substantial decrease in melanoma mortality in an initial pilot program, did not sustain a mortality benefit in the second phase [43]. While participants felt strongly that evidence of mortality benefit was important for acceptability of a screening approach, they acknowledged that the opportunity for a randomised controlled trial to demonstrate this may have passed in the current climate of opportunistic screening and increasing availability of therapies for more advanced disease that are improving survival. Modelling studies have provided evidence that has underpinned changes in the cervical cancer screening program in Australia [44], and was considered a practical way to strengthen the evidence in the absence of trial data. Considering the economic constraints on health care budgets and that cost was seen as a barrier to implementation, highlighting cost savings and lives saved has the potential to engage the government on the value of risk-tailored screening.

Overdiagnosis, referring to the detection of a true melanoma that is slow-growing and would not cause symptoms in a patient's lifetime, is recognised as a problem in the early detection of melanoma and a potential harm in population screening [4, 5, 45]. Overdiagnosis causes harm through the diagnosis itself leading to unnecessary treatment, tests and other healthcare utilisation such as long-term clinical surveillance and fear of recurrence [12]. This

is relevant to melanoma screening in Australia due to the high incidence of in situ melanomas, thin invasive melanomas, and very common keratinocyte carcinomas (basal cell and squamous cell carcinomas) [8, 46]. A risk-tailored screening approach offers a way to address this challenge as reducing screening in low-risk groups minimises overdiagnosis.

In our analysis, TICD determinants of critical importance were: nature of the (professional) behaviour (ie how risk-tailored screening would be delivered); feasibility (the practicality of risk-tailored screening); and skills (needed to adhere to guidelines). These determinants focused almost exclusively on the role of GPs in the delivery of risk-tailored screening, particularly for the initial risk assessment and discussing tailored screening recommendations based on individual risk [34]. GPs in Australia are well placed to deliver screening with many in mainstream general practice or in dedicated skin cancer clinics currently active in melanoma diagnosis and care [47]. However, there was significant concern about the variability of GP skills in conducting skin checks and excisions, and an absence of accreditation indicating proficiency in skin cancer screening. GPs, in particular, expressed the importance of upskilling. Participants differed in their thoughts about who is best placed to conduct screening. Other studies [48] have reported that screening by nurses or clinician assistants may be a more appropriate use of time and skills.

Financial incentives, for example reimbursement for each professional service by allocating a Medicare item number to melanoma screening, were considered essential if GPs were to be engaged in delivering a screening program, although the potential for a conflict of interest in our current health system that rewards more frequent procedures needs to be considered. Many strategies for implementation were linked to determinants related to 'Individual health professional factors' (Domain) and screening delivery and included: 1) Evaluating different risk assessment criteria and models of screening delivery; 2) Developing a risk-tailored screening pathway in conjunction with models of care; and, 3) Upskilling the workforce. Consulting primary health care professionals and the wider healthcare workforce in skill development and program design would be an important step in engaging and building support for implementation. In addition, novel technologies such as total-body 3D imaging machines have the potential to significantly reduce the workload of those delivering screening and standardise diagnosis. Such innovations are the backbone of large-scale Australian research conducted by the Australian Centre for Excellence in Melanoma Imaging and Diagnosis (ACEMID) and the Australian IMAGE trial (Melanoma Surveillance Photography Clinical Trial) that are influencing how lesions will be monitored and melanoma diagnosed in future.

Skin cancer primary prevention interventions are highly cost-effective in Australia [41, 49]. A small number of participants expressed preference for funding for skin cancer prevention programs rather than screening but admitted that skin cancer prevention, on its own, is unable to substantially reduce the increasing burden of melanoma. However, screening was seen as a natural partner to prevention and indeed would provide a substantial opportunity for increasing prevention education.

Another barrier to implementation identified in the study was the potential for poor public health messaging to have a negative impact on community behaviour. Public health messaging that would advocate for less or no screening for those at low-risk may appear to conflict with messages advocating the general benefits of cancer screening and early detection, and participants expressed concern that some at low-risk may opt-out of prevention altogether or feel disappointed to have missed out. Similarly, Dodd et al. reported women's concerns following a change in the screening interval from two to five years in the Australian cervical cancer screening program, and recommended that communication around the natural history, incidence of cancer and how to transition facilitated acceptance of the deintensification [50]. Thus, well-

targeted public health messaging and communication explaining the overall benefits of a risk-tailored melanoma screening program will be crucial for implementation.

Strengths of our study include the large number of interviews with four key groups of experts and leaders with considerable experience in this field, which provide a comprehensive national perspective. Taking an implementation science approach to the collection, analysis and interpretation of the data ensured that our findings and potential implementation strategies are theoretically informed. Our findings contribute to the scarce implementation science literature about risk-stratified melanoma screening by identifying where the barriers lie and what potential strategies are applicable for melanoma. Health policy makers need to ensure that the development of an organised screening program has addressed acceptability if it is to be successfully implemented. A limitation of our study included that the sample was initially identified through knowledge of the authors and may have unintentionally excluded some relevant key informants. However, we sought to overcome any unintentional bias using a snowballing approach. The views and experience of participants were diverse and therefore the results represent a high-level reflection on the topic. Participants offered many suggestions, and not all could be reported in detail in this paper but provide opportunities for further research and engagement. This study was conducted in the context of the Australian mixed public-private healthcare system and a high incidence of melanoma, and so our results may not be generalisable to other countries.

## Summary

In summary, we found clear support in principle from key informants for organised risk-tailored melanoma screening, reporting it to be both acceptable and appropriate. We identified a wide range of implementation strategies including strengthening the evidence for cost-effectiveness, engaging primary health care professionals in planning and development, developing an option or pathway for people at low risk, highlighting potential cost savings and lives saved to government, evaluating different risk assessment criteria and models of screening delivery, developing the risk-tailored screening pathway in conjunction with models of care, and delivering targeted public health messaging. Considerations around risk assessment, policy and modelling the costs of current verses future approaches will be important next steps to inform future possible implementation of risk-tailored population screening for melanoma.

## Supporting information

**S1 File. Interview guide: Risk-tailored melanoma screening study: Stakeholder interviews.**
(PDF)

**S1 Table. Coding framework with domains, relevant determinants, definitions and additional supporting quotes.**
(PDF)

## Acknowledgments

We thank the people, all experts and leaders, who participated in the qualitative interviews. We also thank the Melanoma Screening Modelling and Implementation Project Investigators and collaborators, particularly Associate Professor Craig Sinclair, Professor Karen Canfell, Professor Rachael Morton and Jay Allen for guidance on conceptualisation and recruitment.

## Author Contributions

**Conceptualization:** Kate L. A. Dunlop, Andrea L. Smith, Sanchia Aranda, Joanne Aitken, Caroline G. Watts, Amelia K. Smit, Monika Janda, Anne E. Cust, Nicole M. Rankin.

**Data curation:** Kate L. A. Dunlop.

**Formal analysis:** Kate L. A. Dunlop, Louise A. Keogh, Andrea L. Smith, Sanchia Aranda, Joanne Aitken, Caroline G. Watts, Amelia K. Smit, Monika Janda, Graham J. Mann, Anne E. Cust, Nicole M. Rankin.

**Funding acquisition:** Sanchia Aranda, Joanne Aitken, Anne E. Cust, Nicole M. Rankin.

**Investigation:** Kate L. A. Dunlop.

**Methodology:** Kate L. A. Dunlop, Andrea L. Smith, Sanchia Aranda, Joanne Aitken, Caroline G. Watts, Amelia K. Smit, Monika Janda, Anne E. Cust, Nicole M. Rankin.

**Supervision:** Anne E. Cust, Nicole M. Rankin.

**Writing – original draft:** Kate L. A. Dunlop, Anne E. Cust, Nicole M. Rankin.

**Writing – review & editing:** Kate L. A. Dunlop, Louise A. Keogh, Andrea L. Smith, Sanchia Aranda, Joanne Aitken, Caroline G. Watts, Amelia K. Smit, Monika Janda, Graham J. Mann, Anne E. Cust, Nicole M. Rankin.

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
