## [Decision Letter · Decision Letter 0]

30 Jan 2023

PONE-D-22-30659Acceptability and appropriateness of a risk-tailored organised melanoma screening program: Qualitative interviews with key informantsPLOS ONE

Dear Dr. Dunlop,

Thank you for submitting your manuscript to PLOS ONE. After careful consideration, we feel that it has merit but does not fully meet PLOS ONE’s publication criteria as it currently stands. Therefore, we invite you to submit a revised version of the manuscript that addresses the points raised during the review process.

Please submit your revised manuscript by Mar 16 2023 11:59PM If you will need more time than this to complete your revisions, please reply to this message or contact the journal office at plosone@plos.org. Please include the following items when submitting your revised manuscript:A rebuttal letter that responds to each point raised by the academic editor and reviewer(s). You should upload this letter as a separate file labeled 'Response to Reviewers'.A marked-up copy of your manuscript that highlights changes made to the original version. You should upload this as a separate file labeled 'Revised Manuscript with Track Changes'.An unmarked version of your revised paper without tracked changes. You should upload this as a separate file labeled 'Manuscript'.If applicable, we recommend that you deposit your laboratory protocols in protocols.io to enhance the reproducibility of your results. Protocols.io assigns your protocol its own identifier (DOI) so that it can be cited independently in the future. For instructions see: https://journals.plos.org/plosone/s/submission-guidelines#loc-laboratory-protocols. Additionally, PLOS ONE offers an option for publishing peer-reviewed Lab Protocol articles, which describe protocols hosted on protocols.io. Read more information on sharing protocols at https://plos.org/protocols?utm_medium=editorial-email&utm_source=authorletters&utm_campaign=protocols.

We look forward to receiving your revised manuscript.

Kind regards,

Maryam Farooqui, Ph.D

Academic Editor

PLOS ONE

Journal Requirements:

3. Please upload a new copy of Figure 1 as the detail is not clear. Please follow the link for more information: " ext-link-type="uri" xlink:type="simple">https://blogs.plos.org/plos/2019/06/looking-good-tips-for-creating-your-plos-figures-graphics/"
https://blogs.plos.org/plos/2019/06/looking-good-tips-for-creating-your-plos-figures-graphics/

Reviewers' comments:

Reviewer's Responses to Questions

**Comments to the Author**

1. Is the manuscript technically sound, and do the data support the conclusions?

Reviewer #1: Yes

Reviewer #2: Yes

2. Has the statistical analysis been performed appropriately and rigorously? 

Reviewer #1: N/A

Reviewer #2: Yes

3. Have the authors made all data underlying the findings in their manuscript fully available?

Reviewer #1: Yes

Reviewer #2: Yes

4. Is the manuscript presented in an intelligible fashion and written in standard English?

Reviewer #1: Yes

Reviewer #2: Yes

5. Review Comments to the Author

Reviewer #1: Abstract:

1. Kindly describe methods section in terms of research design, sampling method and data analysis methods employed.

Introduction:

1. The claim in the opening statement is not supported by sufficient evidence. The only reference cited is inconclusive, which doesn’t advocate for “International evidence-based guidelines do not recommend routine screening for melanoma”.

2. Line 92; what are those challenges. Challenges will definitely impact acceptability and of course, appropriateness. Therefore, briefly enlist the main challenges.

3. Lines 96-102; Lack reference

4. Authors have argued that implementation of risk-tailored screening is challenging. I am wondering if they can address the feasibility of intervention first as what is the point of willingness for implementation of an unfeasible intervention. I can understand from Proctor et al’ given framework for ‘outcomes for implementation research’ that acceptability comes before feasibility and cost consideration. But still seems good to anticipate feasibility based on available literature.

5. Overall, introduction section seems reporting available literature rather than building an argument to justify the significance of risk-tailored screening, challenges in its implementation and role of acceptability and appropriateness of a risk-tailored organised melanoma screening program towards its implementation.

Methods, results and discussion are well written and address the concerns highlighted above.

Reviewer #2: 1. Abstract

Suggest changing the pronoun word "we" to "researcher"

Please explain the inclusion and exclusion criteria in the method.

The author should specify what was the 'valuable information' in the conclusion and briefly describe the importance of the 'valuable information"

2. Introduction

The introduction section has a clear statement demonstrating that the focus of the study. The problem definition is stated clearly.

3. Risk-tailored screening, also referred to as ‘risk-stratified screening’ or ‘targeted screening’, tailors screening (e.g. eligibility,frequency, intervals, type of test) to individual risk rather than the mostly one-size-fits-all approach of organised population screening programs.

Please add the reference

Please add citations when necessary.

4. This qualitative study aimed to explore the acceptability and appropriateness of population based, organised risk-tailored melanoma screening from the perspective of key informants.

Please harmonize the statement

5. Methodology

I would like to suggest to the author provide more information on the methodology of the study such as:

-How about the sample size selected?

-The total number of respondents from four expert groups?

-The inclusion and exclusion criteria?

-The semi-structured interview protocol process consisted of what questions?

6. Ethical consideration

Please add the reference number or approval code

7. Table 1

Please add the subtitle in the table

Example:

Frequency(n)

8. Discussion

How do the findings add to the body of scientific knowledge on the particular issue?

Should highlight the implication of the results to health policy.

9.Summary

Please make a subheading for summary

10. Please state the source of the research funding including the grant name and number.

11. References

Please add doi in the article’s references.

Example:

Trager MH, Queen D, Samie FH, Carvajal RD, Bickers DR, Geskin LJ. Advances in Prevention and Surveillance of Cutaneous Malignancies. Am J Med. 2020 Apr;133(4):417-423. doi: 10.1016/j.amjmed.2019.10.008. Epub 2019 Nov 9. PMID: 31712100; PMCID: PMC7709483.

6. PLOS authors have the option to publish the peer review history of their article (what does this mean?). If published, this will include your full peer review and any attached files.

Reviewer #1: No

Reviewer #2: No

---

## [Author Response · Author response to Decision Letter 0]

8 Mar 2023

REVIEWER #1

Abstract: 

1.Kindly describe methods section in terms of research design, sampling method and data analysis methods employed.

AUTHOR RESPONSE: 

We deleted the existing wording in the Methods section (Abstract) and replaced it to include research design, sampling method and data analysis methods as follows:

“This was a qualitative design study using semi-structured interviews. Key informants were purposively selected to ensure expertise in melanoma early detection and screening prioritising senior or executive perspectives. Consumers were expert representatives. Data were analysed deductively using the Tailored Implementation for Chronic Diseases (TICD) checklist.” [p2, Line 50-54 tracked manuscript]

Introduction:

1.The claim in the opening statement is not supported by sufficient evidence. The only reference cited is inconclusive, which doesn’t advocate for “International evidence-based guidelines do not recommend routine screening for melanoma”.

AUTHOR RESPONSE: 

For clarity we have included a new opening statement, revised the previous opening statement to align correctly with the reference and added additional references as follows: 

“Early detection of melanoma is associated with better survival [1]. However, international evidence-based guidelines report there is insufficient evidence to recommend for or against routine screening for melanoma in the asymptomatic population [2]” due to insufficient evidence that screening using current approaches reduces mortality [3, 4], concern about the potential harms associated with the overdiagnosis of slowly-progressive lesions [4-6] and that costs may outweigh benefits. [p4, Line 74-80 tracked manuscript]

2. Line 92; What are those challenges. Challenges will definitely impact acceptability and of course, appropriateness. Therefore, briefly enlist the main challenges.

AUTHOR RESPONSE:

We have expanded on the challenges with additional wording and references for clarity as follows:

 “Moving from opportunistic screening to an organised risk-tailored approach to melanoma screening will be challenging, as it will require changes to practice and policy [21, 24]. Nevertheless, several countries are moving forward with implementing risk-stratified screening as part of their long-term cancer plans. [25-27]” [p5,102-105 tracked manuscript]

“This would include redistributing health service resources towards higher-risk individuals to screen more frequently whereas those at lower risk to screen less frequently or perhaps not all. [p5, Line 111-114 tracked manuscript]

“Integrating risk factors such as lifestyle behaviours, environmental exposures and personal genomic risk information in addition to traditional risk factors to determine screening eligibility will also bring challenges in communication[26, 27]. These potentially significant policy and practice changes… [p5, Line 111-117 tracked manuscript]

3. Line 96-102; Lack of reference.

AUTHOR RESPONSE:

References have been added to support existing information added, related to challenges as above:

Line: 96-97: Reference added [p5, Line 111 tracked manuscript]

Line: 100-102: Reference added [p5, Line 120 tracked manuscript]

Line: 105-107: References were moved to reflect findings and an additional reference added:

Acceptability of risk-tailored screening programs by key stakeholders has been explored primarily in breast cancer reporting a high level of acceptability [28, 29] but with some reluctance to stop organised screening for those at low risk [30, 31] [p5, Line 119 and 120 tracked manuscript]. To our knowledge, no studies have explored stakeholders’ views of the acceptability and appropriateness of risk-tailored melanoma screening [37]. [p5-6, Line 121-125 tracked manuscript]

4. Authors have argued that implementation of risk-tailored screening is challenging. I am wondering if they can address the feasibility of intervention first as what is the point of willingness for implementation of an unfeasible intervention. I can understand from Proctor et al’ given framework for ‘outcomes for implementation research’ that acceptability comes before feasibility and cost consideration. But still seems good to anticipate feasibility based on available literature.

5.Overall, introduction section seems reporting available literature rather than building an argument to justify the significance of risk-tailored screening, challenges in its implementation and role of acceptability and appropriateness of a risk-tailored organised melanoma screening program towards its implementation. 

Methods, results and discussion are well written and address the concerns highlighted above.

AUTHOR RESPONSE:

We thank the reviewer for this observation and guidance. We have revised the wording in the introduction to address the feasibility of the intervention (risk-tailored organised melanoma screening) and to help build an argument to justify its significance as follows:

We revised the second paragraph of the introduction to read “A risk-tailored approach to screening for melanoma offers a promising way forward [3, 13-15] and melanoma risk prediction tools validated for the Australian population are available [16-18]. New technologies such as surveillance photography, teledermatology and artificial intelligence, have the potential to support clinicians in screening[19].” [p4, Line 90-94 tracked manuscript]

We also added the sentence: 

‘This approach will be timely while evidence for this new paradigm in melanoma screening is further developed.” [p6, Line 128-129 tracked manuscript]

REVIEWER #2

General comments

Congratulations on the submitted manuscript. The topic is timely and will be of interest to the readers of the journal. However, few changes are suggested to improve the clarity of this manuscript. I have several recommendations and questions about the manuscript.

1. Abstract: Suggest changing the pronoun word "we" to "researcher"

AUTHOR RESPONSE:

‘We’ has been changed to ‘this study’ [p2, Line 41]

‘We identified’ has been replaced by ‘A’ and ‘were identified’ added [p2, Line 62-63 revised manuscript]

Changed ‘models of’ to ‘models’ to reduce word count [p3, Line 65]

Please explain the inclusion and exclusion criteria in the method.

AUTHOR RESPONSE:

We revised the Methods section (Abstract) in response to Reviewer 1 comments. For clarity we have deleted “with policy makers, consumers with lived experience of melanoma or community advocates, healthy professionals and researchers” and added 

‘Key informants were purposively selected to ensure expertise in melanoma early detection and screening, prioritising senior or executive perspectives. Consumers were expert representatives.’ [p2, Line 50-53]

The author should specify what was the 'valuable information' in the conclusion and briefly describe the importance of the 'valuable information"

AUTHOR RESPONSE:

For clarity, we added:

“Key informants were supportive in principle of risk-tailored melanoma screening, highlighting important next steps.” [p3, Line 68-69 tracked manuscript]

and simplified the final following sentence to read: 

“Considerations around risk assessment, policy and modelling the costs of current verses future approaches will help inform possible future implementation of risk-tailored population screening for melanoma”.

2.Introduction: The introduction section has a clear statement demonstrating the focus of the study. The problem definition is stated clearly.

AUTHOR RESPONSE: No changes needed.

3. Risk-tailored screening, also referred to as ‘risk-stratified screening’ or ‘targeted screening’, tailors screening (e.g. eligibility, frequency, intervals, type of test) to individual risk rather than the mostly one-size-fits-all approach of organised population screening programs.

- Please add the REFERENCE

Please add citations when necessary

AUTHOR RESPONSE:

Two references [20,21] have been added to support this claim. [p4, Line 97 tracked manuscript]

4.This qualitative study aimed to explore the acceptability and appropriateness of population based, organised risk-tailored melanoma screening from the perspective of key informants.

Please harmonize the statement

AUTHOR RESPONSE:

We have revised the statement as follows:

“This study aimed to explore the views of key informants in Australia on the acceptability and appropriateness of risk-tailored organised screening for melanoma.” [p6, Line 133-135 tracked manuscript]

5. Methodology: I would like to suggest to the author provide more information on the methodology of the study such as:

-How about the sample size selected?

-The total number of respondents from four expert groups?

-The inclusion and exclusion criteria?

-The semi-structured interview protocol process consisted of what questions?

AUTHOR RESPONSE:

-To explain sample size selected we included: “Recruitment continued until no new information was generated (i.e., reached data saturation).” [p7, Line 156-157 tracked manuscript]

-The total number of respondents from four expert groups are included in Table 1 and in the Abstract.

-To further clarify inclusion and exclusion criteria we added:

“Key informants were included if they were recognised as a leader in their field, a clinical/policy expert or experienced representative of melanoma patient or community groups.” [p6, Line 144-146 tracked manuscript]

“Participants were excluded if they did not hold a current position associated with the key organisations.” [p7,149-150 tracked manuscript]

-The semi-structured interview guide with questions used in the interview process is included in S1 File. In addition, the following sentence included in Data collection (Methodology) highlights the main focus of interview questions: “Interview questions explored beliefs and perceptions about melanoma early detection and screening in Australia, the concept of risk-tailored screening, evidence for implementation and the key determinants (barriers and facilitators).” [p7, Line 174-176)

6. Ethical Consideration: Please add the reference number or approval code

AUTHOR RESPONSE:

The following statement is included on page 7, Line 157-158. “This research was approved by The University of Sydney Human Research Ethics Committee (Project 2021/253).”

7. Table 1: Please add the subtitle in the table

Example:

Frequency(n)

AUTHOR RESPONSE:

Subtitles have been added to Table 1 [p10, Line 224]

8. Discussion: How do the findings add to the body of scientific knowledge on the particular issue? Should highlight the implication of the results to health policy

AUTHOR RESPONSE:

As suggested, we have included the following sentences:

“Our findings contribute to the scarce implementation science literature on risk-stratified melanoma screening by identifying where the barriers lie and what potential strategies are applicable for melanoma. Health policy makers need to ensure that the development of an organised screening program has addressed acceptability if it is to be successfully implemented.” [p26, Line 604-609 tracked manuscript] 

9. Summary. Please make a subheading for summary

AUTHOR RESPONSE:

The subheading “Summary” has been added. [p27, Line 619 tracked manuscript]

10. Please state the source of the research funding including the grant name and number.

AUTHOR RESPONSE:

The source of research funding has been submitted as part of the submission information as required by PLOS ONE submission guidelines and has been updated for clarity to read:

Funding: This study has received project grant funding from the National Health and Medical Research Council (NHMRC #1165936 and #2009923). KLAD receives an NHMRC (Australia) Postgraduate Research Scholarship and NHMRC Supplementary Scholarship The University of Sydney 2021-2022 and The Erik Mather PhD Scholarship (Melanoma Institute Australia). AEC receives a NHMRC Investigator Grant (2008454). The funders had no role in study design, data collection and analysis, decision to publish, or preparation of the manuscript.

11. References: Please add doi in the article’s references. Example:

Trager MH, Queen D, Samie FH, Carvajal RD, Bickers DR, Geskin LJ. Advances in Prevention and Surveillance of Cutaneous Malignancies. Am J Med. 2020 Apr;133(4):417-423. doi: 10.1016/j.amjmed.2019.10.008. Epub 2019 Nov 9. PMID: 31712100; PMCID: PMC7709483.

AUTHOR RESPONSE:

Thank you for drawing our attention to the need for a doi. We have amended the Reference list to include this information where available.

ADDITIONAL AUTHOR CHANGE:

Results: On review, we changed the word ‘disorganised’ to ‘fragmented’ to better reflect the associated quote [p12, Line 248 tracked manuscript].

---

## [Decision Letter · Decision Letter 1]

8 Jun 2023

Acceptability and appropriateness of a risk-tailored organised melanoma screening program: Qualitative interviews with key informants

PONE-D-22-30659R1

Dear Dr. Dunlop,

We’re pleased to inform you that your manuscript has been judged scientifically suitable for publication and will be formally accepted for publication once it meets all outstanding technical requirements.

Kind regards,

Jianhong Zhou

Staff Editor

PLOS ONE

Additional Editor Comments (optional):

Reviewers' comments:

Reviewer's Responses to Questions

**Comments to the Author**

1. If the authors have adequately addressed your comments raised in a previous round of review and you feel that this manuscript is now acceptable for publication, you may indicate that here to bypass the “Comments to the Author” section, enter your conflict of interest statement in the “Confidential to Editor” section, and submit your "Accept" recommendation.

Reviewer #1: All comments have been addressed

Reviewer #2: All comments have been addressed

2. Is the manuscript technically sound, and do the data support the conclusions?

Reviewer #1: Yes

Reviewer #2: Yes

3. Has the statistical analysis been performed appropriately and rigorously? 

Reviewer #1: N/A

Reviewer #2: Yes

4. Have the authors made all data underlying the findings in their manuscript fully available?

Reviewer #1: Yes

Reviewer #2: Yes

5. Is the manuscript presented in an intelligible fashion and written in standard English?

Reviewer #1: Yes

Reviewer #2: Yes

6. Review Comments to the Author

Reviewer #1: Thank you for addressing comments. I hope it come out as a valuable contribution to the body of knowledge.

Reviewer #2: Congratulations to the authors. All corrections have been made based on the reviewers comments. This finding will contribute to the valuable literature in risk-tailored population screening for melanoma.

7. PLOS authors have the option to publish the peer review history of their article (what does this mean?). If published, this will include your full peer review and any attached files.

Reviewer #1: **Yes: **Dr. Ahlam Sundus

Reviewer #2: No

---

## [Editor Report · Acceptance letter]

1 Dec 2023

PONE-D-22-30659R1 

Acceptability and appropriateness of a risk-tailored organised melanoma screening program: Qualitative interviews with key informants 

Dear Dr. Dunlop:

I'm pleased to inform you that your manuscript has been deemed suitable for publication in PLOS ONE. Congratulations! Your manuscript is now with our production department. 

Kind regards, 

on behalf of

Jianhong Zhou 

Staff Editor

PLOS ONE